# Biomarkers for Serious Bacterial Infections in Febrile Children

**DOI:** 10.3390/biom14010097

**Published:** 2024-01-12

**Authors:** Luca Bernardi, Gianluca Bossù, Giulia Dal Canto, Giuliana Giannì, Susanna Esposito

**Affiliations:** Pediatric Clinic, Department of Medicine and Surgery, University of Parma, 43126 Parma, Italy; bernardi.luca.91@gmail.com (L.B.); gianluca.bossu@unipr.it (G.B.); giu.dalcanto@gmail.it (G.D.C.); giuliana.gianni@unipr.it (G.G.)

**Keywords:** biomarker, C reactive protein, interleukin, presepsin, procalcitonin, serious bacterial infection

## Abstract

Febrile infections in children are a common cause of presentation to the emergency department (ED). While viral infections are usually self-limiting, sometimes bacterial illnesses may lead to sepsis and severe complications. Inflammatory biomarkers such as C reactive protein (CRP) and procalcitonin are usually the first blood exams performed in the ED to differentiate bacterial and viral infections; nowadays, a better understanding of immunochemical pathways has led to the discovery of new and more specific biomarkers that could play a role in the emergency setting. The aim of this narrative review is to provide the most recent evidence on biomarkers and predictor models, combining them for serious bacterial infection (SBI) diagnosis in febrile children. Literature analysis shows that inflammatory response is a complex mechanism in which many biochemical and immunological factors contribute to the host response in SBI. CRP and procalcitonin still represent the most used biomarkers in the pediatric ED for the diagnosis of SBI. Their sensibility and sensitivity increase when combined, and for this reason, it is reasonable to take them both into consideration in the evaluation of febrile children. The potential of machine learning tools, which represent a real novelty in medical practice, in conjunction with routine clinical and biological information, may improve the accuracy of diagnosis and target therapeutic options in SBI. However, studies on this matter are not yet validated in younger populations, making their relevance in pediatric precision medicine still uncertain. More data from further research are needed to improve clinical practice and decision making using these new technologies.

## 1. Introduction

Febrile illness is a common pediatric presentation, accounting for 14–20% of attendance to the pediatric emergency department (ED) [1,2,3]. It is frequently the expression of an underlying infection, and a large proportion of these cases are viral in origin, with a benign and self-limiting course. However, in a certain percentage of children, fever can be the manifestation of a bacterial infection, which can be serious (such as septicemia, meningitis, confirmed appendicitis, pneumonia, osteomyelitis, cellulitis, or a complicated urinary tract infection), and the consequences of missing the diagnosis can potentially be catastrophic. Neonates and infants are at a higher risk of serious bacterial infection (SBI) and may not display the same clinical features of infection and sepsis as older children, making the assessment more challenging. Distinguishing between a SBI requiring antibiotics and a viral infection is mostly a clinical decision, and international evidence-based guidelines are a useful tool in that scenario [4,5]. In addition, co-infections between viruses (i.e., influenza, respiratory syncytial virus, rhinovirus, metapneumovirus) are common at the pediatric age, and in some cases, symptoms of co-infections are difficult to distinguish from viral infections alone and in other cases from SBI [6,7,8].

Inflammatory biomarkers are usually the first investigations required in the ED, which help differentiate bacterial from nonbacterial infections in febrile infants, and their result influences the subsequent case management. Stol et al. summarized the properties of the perfect biomarker for infections: has a positive test result in infected patients, has a negative test result in patients without an infection, distinguishes etiology, is independent of comorbidities, is a predictor of severity, is a predictor of outcome, is a quick and easy test with a small variation coefficient, and is affordable [9]. Currently, no single biomarker has sufficient diagnostic accuracy to satisfy all these properties, and the clinical context remains vital in the diagnostic and therapeutic process. In this review, we summarize the current knowledge about the role of biomarkers for SBI in children presenting with fever in the ED, and we discuss the future perspective in this field. To this end, we conducted electronic research in the PubMed database from September 2018 to September 2023, using “sepsis” OR “severe bacterial infection” AND “infant” OR “children” OR “pediatric” OR “paediatric” OR “biomarker” OR “blood culture” OR “blood cell count” OR “neutrophil count” OR “ANC” OR “C-reactive protein“ OR “CRP” OR “procalcitonin” OR “PCT” OR “inflammatory markers” OR “cytokine“ OR “IL-2” OR “IL-6” OR “IL-10” OR “IL-27” OR “soluble triggering receptor” OR “sTREM-1” OR “platelet” OR “TRAIL” OR “IP-10” OR “presepsin” as keywords. Guidelines and position papers published from 2003 to 2018 were also considered. Only articles written in English were selected, and a manual search of the references of eligible articles was made. We did not include manuscripts on biomarkers in neonatal sepsis because we have already analyzed this topic in a previous publication [10].

## 2. Hematological Biomarkers

### 2.1. White Blood Cell Count (WBC) and Absolute Neutrophil Count (ANC)

WBC and ANC have been widely used worldwide as predictors of SBI in febrile children. During a bacterial infection, neutrophils are rapidly recruited to infection sites where they evoke an immune response, bind, and ingest microorganisms by phagocytosis and kill microbes [11]. A larger number of neutrophils are consumed at the site of a SBI, and they continue to be supplied to the infected site from the bone marrow via the bloodstream [12]. Therefore, dynamic changes occur in WBC and ANC that may reflect the real-time condition of a patient with a bacterial infection. However, a recent systematic review and meta-analysis of diagnostic studies showed that the WBC offers a low sensitivity (58%) and a specificity of 73%, or lower if comparted to procalcitonin (PCT) and C-reactive protein (CRP) analysis [12]. Similarly, a study that compared the WBC, ANC, and CRP in relation to the onset of fever found that CRP had a better sensitivity and specificity than either WBC or ANC, regardless of the duration of fever. Interestingly, in this study, all biomarkers performed better with a duration of fever of >12 h [13]. Van den Bruel et al. investigated the diagnostic value of laboratory tests for the diagnosis of SBI in febrile children in ambulatory settings and found that the WBC probably provides some diagnostic value in ruling in serious infections, but less than PCT and PCR, and has no value in ruling it out [14].

### 2.2. Platelet Indices

Studies have identified platelets as one of the first-line indicators in response to pathogens with participation to phagocytosis through proteins from their granules. Different platelet indices, such as PNLR (platelet-to-neutrophil/lymphocyte ratio), PNR (platelet-to-neutrophil ratio) and secreted proteins, such as sP-selectin, CXCL4, CXCL7, and serotonin, have been studied as markers to discriminate viral and bacterial infection pathogenesis [15].

Considering children who present in the ED with early onset of fever (<12 h), a higher PNLR value has been observed in those suffering from bacterial infections [16]. sP-selectin in the ED may discriminate between septic and non-septic patients [16].

CXCL7 has a valid specificity and sensibility in detecting early signs of sepsis and excluding other causes of SIRS. CXCL7 and sP-selectin, alone and combined, are statistically significant to discriminate sepsis and bacterial infections from other diseases [17]. In pediatric patients in whom an acute infectious event is suspected, CXCL4 and serotonin levels are not indicative in discriminating the etiology of the event in progress; CXCL4 has a role during the viral response, and its elevation in the blood stream is not significant in patients with sepsis or bacterial infections [18]. However, the values are not yet standardized in the pediatric population, and more studies are necessary to confirm normal values in healthy children and in different clinical conditions, i.e., chronic inflammation, trauma, and acute infection [19]. CXCL7 and sP-selectin are promising for the future, and the aim is to understand how to correlate early signs of infection to these biomarkers’ levels, improving the recognition of a bacterial infection from a viral one and contextually SBI [20].

## 3. Inflammatory Biomarkers

### 3.1. C-Reactive Protein (CRP)

CRP is currently one of the most frequently used biomarkers for infection in the ED worldwide [21]. It is a short pentraxin, which is synthesized in the liver following stimulation by cytokines (IL-1beta, IL-6, and TNF-alfa) within 4–6 h after tissue injury, doubling every 8 h, and peaking at 36–50 h [22,23]. CRP plays an important role in host defense through complement activation via the classic pathway, modulation of the function of phagocytic cells, and an increase in cell-mediated cytotoxicity [24].

A rise of CRP levels can be caused by conditions other than infections, for example, trauma, malignancy, rheumatologic disorders, burns, pancreatitis, and periodic fever syndromes, and CRP values should be interpreted cautiously in these cases [25]. On the contrary, suppressed levels of CRP can be present in liver failure and immunocompromised patients [26]. Nevertheless, several studies demonstrate the utility of CRP for early identification of febrile children at risk for SBI [14].

A recent systematic review and meta-analysis evaluated the diagnostic value of CRP for early identification of young children at risk for SBI among those presenting with fever without source and found that overall sensitivity was 0.74 (95% confidence interval [CI], 0.65 to 0.82) and overall specificity was 0.76 (95% CI, 0.70 to 0.81) [12].

A crucial dilemma in clinical practice is the threshold to use for the identification of SBI. A very low cut-off value will be very sensitive but poorly specific, and a very high cut-off will be specific but poorly sensitive [27]. In a recent study by Verbakel et al., the cut-off value of 75 mg/L was suggested as highlighting those children at a greater risk of SBI, and a CRP cut-off of 20 mg/L was suggested as being useful in identifying children at a low risk of SBI [28]. The CRP value must be interpreted with caution when fever has been present <12 h based on the kinetics of this biological marker [13].

Studies showed that high levels of CRP and PCT are strongly predictive of SBI in children with fever, independent of duration of disease; on the contrary, low CRP levels should not be used to rule out or confirm SBI in children with a short duration of fever, and PCT seems superior to CRP in detecting SBI at an earlier stage of the disease [10,13,29].

Neonates and infants < 3 months deserve specific considerations [30]. A large multicentered European study of over 2000 infants under 3 months of age admitted to a pediatric ED with fever without source found that CRP was a poor predictor of SBI [30]. A 70 mg/L cut-off had a specificity of 93.8%, but sensitivity of only 69.6%. In this study, the CRP value was higher than the WBC and ANC in detecting bacteremia, but the most accurate predictor of SBI was appearing unwell [30]. Similarly, one large multicentered American study of suspected sepsis in neonates found the initial CRP value to be poorly sensitive for SBI [31]. However, they reported that an elevated CRP > 10 mg/L at 24–48 h after presentation demonstrated a 97.6% and 94.4% sensitivity for proven (culture positive) or probable (clinical features but no positive cultures) bacterial infection, respectively, making serial CRP measurements more accurate in diagnosing SBI in neonates.

### 3.2. Procalcitonin

Procalcitonin (PCT) is a 116—amino acid protein precursor for calcitonin produced by parafollicular cells [32]. In normal conditions, serum levels of PCT are lower than 0.05 ng/mL, while during SBI, they can increase up to 700 ng/L [33]. During SBI, the site of PCT production is not limited to the neuroendocrine cells. The release of PCT is induced by increasing the CALC1 gene expression in parenchymal cells throughout the body, triggered by endotoxin or by humoral factors, i.e., IL-1, TNF- alfa, and IL-6 [34,35].

PCT concentrations increase more rapidly than CRP levels in patients with SBI. PCT levels begin to increase at 2 h from the onset of infection and reach a serum peak at 24 to 36 h [36]. For this reason, PCT has been shown to be a superior biomarker as compared with CRP for detecting SBI in the ED [37]. However, the specificity for detecting SBI is limited, especially for infants < 3 months [10,38].

In a consistent meta-analysis, England et al. showed that serum PCT concentrations < 0.3 ng/mL identified a population of febrile infants < 91 days of age at low risk for SBI [39]. They concluded that the serum PCT concentration alone is a poorer predictor of SBI and may be used in combination with clinical valuation.

A meta-analysis to investigate the diagnostic accuracy of PCT as an early biomarker of sepsis was performed, including 1408 patients (1086 neonates and 322 children) [40]. In the neonatal group, PCT showed a sensitivity of 85% (95% CI, 76% to 90%) and a specificity of 54% (95% CI, 38% to 70%) at the PCT cut-off of 2.0–2.5 ng/mL. In the pediatric group, it was not possible to undertake a pooled analysis at the PCT cut-off of 2.0–2.5 ng/mL due to the paucity of the studies [36]. In a recent prospective multi-center cohort study, Waterfield et al. revealed no difference and only a moderate accuracy for PCT and CRP in detecting SBI in the ED, reporting that the area under the curve was identical at 0.70 [41].

The diagnostic power of PCT in the pediatric intensive care unit (PICU) is uncertain. PCT adequately predicted SBI in a heterogeneous PICU population, with a PCT of ≥1.28 ng/mL as the ideal threshold for detection of SBI, as reported in a recent retrospective cohort study [42]. Another retrospective study performed in the PICU identified a PCT value of ≥1 ng/mL as able to predict SBI with a sensitivity of 70% and a specificity of 68% [43]. In a retrospective observational study involving 646 critically ill children, Lautz et al. found that a peak blood PCT measured within 48 h of PICU admission was not superior to CRP in differentiating SBI from viral illness and sterile inflammation, raising doubts about the right timing to perform PCT in the PICU [44]. Zeng et al., in a recent retrospective analysis, found that PCT alone was not better able to diagnose the hyperinflammatory state than CRP in the PICU [45]. Furthermore, when both biomarkers were simultaneously elevated, the diagnostic specificity of SBI increased.

### 3.3. Cytokines and Chemokines

Pattern recognition receptors (PRR) not only recognize pathogen-associated molecular markers (PAMPs, e.g., endo- and exotoxins, DNA, lipids) of foreign invaders, but also endogenous host-derived danger signals (damage-associated molecular patterns, DAMPs) [46]. The interaction of Toll-like receptors (TLRs) located on the membrane surfaces of antigen-presenting cells (APCs) and monocytes with PAMPs or DAMPs results in the initiation of signaling cascades and the expression of genes involved in inflammation, adaptive immunity, and cellular metabolism. This leads to the expression of so-called “early activation genes” and to the release of cytokines (e.g., IFN-γ, IL-1, IL-6, IL-8, IL-12) and components of the complement and coagulation systems [47]. This systemic increase in pro- and anti-inflammatory cytokines in the early phase is considered the classic hallmark of SBI. The proinflammatory components cause inflammation, which, if systemic, can lead to progressive tissue damage and to organ dysfunction. Concomitant immune suppression caused by downregulation of activating cell surface molecules increases apoptosis of immune cells, and depletion of T cells leads to “immune paralysis” in later stages of the disease course, making the organism susceptible to nosocomial infections, opportunistic pathogens, and viral reactivation [48].

Because of the early involvement in the host immune response to infections, cytokines and chemokines have been considered as promising biomarkers of SBI, especially in recent years, when most problems of their detection in blood samples have been solved. Moreover, as CRP and PCT production depends on cytokine release, it was thought that the measure of cytokines could offer an earlier and more effective evaluation of sepsis development compared to the traditionally used biomarkers [10]. Unfortunately, not all the expected benefits have materialized.

#### 3.3.1. Interleukines (IL)

IL-2 is indicated as the most specific biomarker in patients with SBI, with low sensitivity and moderate specificity (54% and 86%, respectively) [49]. The poor predictive accuracy of this molecule does not permit it to be considered as an optimal biomarker for sepsis in clinical practice.

IL-6 has been studied for its role in systemic inflammation. It is described as an acute phase pro-inflammatory cytokine, which increases its blood level within the first 6 h, earlier than CRP, during bacterial infections [50]. It turns out to be useful in predicting SBI diagnosis in children with fever without an apparent source [51]. In a large prospective study, even if the blood level of IL-6 was higher in septic children, the difference between the septic and non-septic group was not statistically significant [52]. Comparing blood draws collected at different arrival times, the sensitiveness decreases as the hours pass from the onset of the fever. Although pediatric data are few, evidence on the role of IL-6 in neonates with sepsis is promising [53,54]. IL-6 appeared as an early marker of neonatal sepsis, even if its levels tend to normalize during the development of infection, increasing false-negative findings [55,56].

The key role of increasing levels of IL-10 in the anti-inflammatory response causes worse outcomes in oncologic neutropenic patients with sepsis [57]. In recent findings, IL-10 appeared with a high specificity and moderate sensitivity. While IL-6 decreases quickly in the first 12 h from the onset of the blood infection, IL-10 tends to persist for longer during the septic state and performs as a valuable diagnostic biomarker [57].

However, many authors declared the superiority of combinations of blood biomarkers over individual tests in the differential diagnosis of infection etiology [58,59]. It has been described that the combination of WBC, ANC, CRP, IL-2, and IL-6 increase sensitivity to 96%, specificity of 81%, and a large AUC 0.942 (CI 95%, 0.859 to 0984) in differentiating bacterial pathogenesis [45]. Similarly, matching CRP with IL-10 levels, the clinician obtained a higher discriminative ability in the etiology of infection (specificity from 77% to 98%, sensitivity 75%) [60].

Finally, recent preliminary studies have shown promising results on the specificity of IL-27 in the early prediction of SBI in critical pediatric patients. Using a large genome-wide expression database of critical children in the pediatric ED, predictor genes coding for the IL-27 protein were described; in particular, EB13, a subunit of IL-27, appeared to have a high predictive role for bacterial infections (more than 90%) [61]. In comparison to PCT, IL-27 performed better in discriminating bacterial from viral infections. These findings, although preliminary, lead to considering IL-27 as an effective biomarker in bacterial sepsis, exhibiting a specificity of 95% in detection of infection. A CART-generated algorithm including IL-27, PCT, and immune status led to an undisputed improvement in predictive value, statistically improved from either IL-27 or PCT alone [62].

#### 3.3.2. TRAIL and IP-10

Tumor necrosis factor-related apoptosis-inducing ligand (TRAIL) is a type II transmembrane protein belonging to the TNF superfamily, which is involved in infection control and in the regulation of both innate and adaptive immune responses [63]. TRAIL is involved in sepsis by inducing apoptosis of inflammatory cells and downregulating inflammation [64]. Many authors have explored the association between soluble TRAIL (sTRAIL) levels in septic patients and the risk of mortality: low sTRAIL levels seem to be associated with a high risk of mortality, with survivor patients who had significantly higher levels of sTRAIL than non-survivors [65,66,67,68].

IP-10 (i.e., interferon-gamma-inducible protein 10) is a chemokine that is expressed by antigen-presenting cells in response to IFN-γ and attracts activated T-cells to the foci of inflammation [69]. This biomarker plays a role in the response to bacterial infections, particularly in the diagnosis and management of urinary tract infections, tuberculosis, and inflammatory diseases such as Kawasaki disease [70,71,72].

Van Houten et al. found that with an assay combining three biomarkers, i.e., TRAIL, IP-10, and CRP, it is possible to distinguish bacterial from viral infections in febrile children with a sensitivity of 86.7% and a specificity of 91.1% [73]. In a proteomics-based study focusing on the host immune response, Oved et al. demonstrated that the combination of these three biomarkers showed a better performance compared to different combinations of routine biomarkers of inflammation in patients suffering from infectious diseases or from fever with unknown disease [74]. Papan et al., in a multinational prospective cohort study, validated the diagnostic performance of the novel host-response-based signature comprising TRAIL, IP-10, and CRP in a broad cohort of pediatric patients with respiratory tract infection or fever without source, demonstrating its capability to support the diagnosis of viral etiology and reducing the prescription of antibiotics [75]. Figure 1 shows how the novel host-response-based signature comprising TRAIL, IP-10, and CRP works.

## 4. Cell Adhesion Molecules

Several cell adhesion molecules, including presepsin, cluster differentiation molecule-64 (CD64), soluble trigger receptor expressed on myeloid cell-1 (sTREM1), and pentraxin3, were tentatively used to differentiate septic children from non-septic ones [76]. However, only presepsin and sTREM1 were used in a number of studies that were useful for drawing some conclusion regarding their role in this regard.

### 4.1. Presepsin

Presepsin (sCD14-ST) is a protein related to the cleavage of CD14, a soluble form of lipopolysaccharide (LPS) receptor, which recognizes pathogen-associated molecular patterns (PAMPs) and triggers the innate immune response [77]. This explains its specific elevation in bacterial infections, in which the underlying pathogenetic mechanism is expressed through the action of LPS.

Presepsin seems to have good specificity and sensitivity in sepsis and correlates to in-hospital mortality in patients with sepsis and septic shock, with a diagnostic potential that can increase if it is combined with clinical scores [78]. During the bacterial infectious state, the concentration in absolute value increases within 2 h. Different studies reported that presepsin is the only biomarker that, if it remains elevated in a patient with a SBI, it could be associated with a higher risk of mortality throughout the follow-up period [79]. However, despite the literature supporting its potential role in the ED and in the intensive care setting, some studies do not indicate a superiority of presepsin compared to other biomarkers in terms of sensitivity and specificity [80].

In neonatal sepsis, presepsin offers the advantage of identifying culture-negative sepsis, with the possibility of early initiation of antibiotic therapy [81]. Meanwhile, presepsin excludes the diagnosis of sepsis in newborns not likely to be affected, reducing the misuse of antibiotics, minimizing hospital stays, and avoiding selection pressure for resistant strains [82]. Levels of presepsin are significantly higher in neonates with sepsis than in healthy ones, and they increased earlier than PCT or CRP; the rise in blood values of CRP and PCT is similarly high during the early phase of infection, but presepsin alone decreases with antibiotic treatment [83,84].

The use in clinical practice of a combination model including presepsin in addition to CRP and PCT may be useful for the early detection of SBI in children with fever admitted to the ED and for monitoring the response to therapy.

### 4.2. STREM-2

Triggering receptor expressed on myeloid cells 1 (TREM-1) is an innate immune receptor that plays an important role in the amplification of the innate immune response to infection by stimulating the release of pro-inflammatory cytokines [85]. sTREM-1 is released from monocytes/macrophages and neutrophils during activation. The presence of bacterial infection increases sTREM-1 levels [86]. The soluble form of this receptor, sTREM-1, is released from the cell membrane and secreted into the circulation during infection [87]. Previous literature data show that sTREM-1 could be used as a marker of severity and outcome in septic neonates [88,89], while its diagnostic potential in pediatric patients older than one month seems to be moderate [90]. Systematic reviews and meta-analysis have recently evaluated the potential role of sTREM as a support in SBI diagnosis. However, low sensitivity and moderate specificity for sTREM-1 in distinguishing bacterial or viral etiology of infections were reported [86,91].

## 5. Future Perspective

In pediatric patients, SBI is defined as the presence of the systemic inflammatory response syndrome (SIRS) during evidence of an infection based on pathogen identification in the bloodstream or by the presence of symptoms directly linked to a high probability of systemic bacterial infection [92]. Early recognition of sepsis in children based on these definitions is often problematic, since blood cultures often provide false negative results, and clinical symptoms are very unspecific, so emergency-setting management results in a delay of an adequate antimicrobial administration [93,94]. In addition, the time to positivity of blood cultures should be considered. It is well known that in patients with a central line, an earlier positivity of central compared with peripheral venous-blood cultures is observed [95,96]. Regarding otherwise healthy children, it is important to remember that more than 85% of all cultures containing pathogens are detected in samples obtained from peripheral blood within the first 24 h of incubation [97,98,99,100,101,102]. Continuously monitoring blood culture systems allows for early identification, taking into account that a short time to positivity is a reliable marker for patient outcomes in certain bacterial species [103].

Nowadays, it is clear that a combination of several SBI biomarkers instead of using one at a time can improve the accuracy of SBI identification by unifying them into one diagnostic model/algorithm, as seen in adult patients [104]. Researchers also agree on the fact that crossing sepsis biomarkers with clinical and epidemiological information further optimizes accuracy. A retrospective cohort study aimed to evaluate the performance of a two-step decision support algorithm based on an electronic health record best-practice alert (BPA) with age-adjusted vital sign ranges and a physician screen [105]. The BPAs rely on the presence of clinical markers of possible infection and incorporate patient risk factors, using demographic data, prior surgeries, or the patient’s problem list and/or medication list to recognize three different types of SBI risks, stratified by the severity of the patient’s underlying disease, with results that seem less specific in adults compared with children [106].

A German group has tried to develop and validate a diagnostic model for the discrimination of pediatric SBI and non-infectious SIRS, which could be set as an algorithm immediately ready for clinical practice [107]. Starting from a secondary analysis of a randomized controlled trial, they created a model including four clinical (length of PICU stay until onset of non-infectious SIRS/SBI, central line, core temperature, number of non-infectious SIRS/SBI episodes prior to diagnosis) and four laboratory parameters (interleukin-6, platelet count, procalcitonin, CRP), through a data-driven analysis approach. The authors stated that the model could potentially reduce antibiotic treatment by 30% in non-infectious SIRS, emphasizing the importance of combining biomarkers and clinical parameters [107].

On this matter, there have been advances in the use of data-driven techniques to improve recognition of early signs of SBI: prediction models have been studied to obtain with machine learning a class of mathematical methods that attempt to generate knowledge and insight from large datasets [108]. Machine learning techniques have also been useful for the evaluation of inflammatory sub-phenotypes based on measurements of panels of inflammatory mediators either alone or in conjunction with clinical variables.

Considering both routine variables and inflammatory biomarkers in patients affected by acute respiratory distress syndrome (ARDS), a common complication of SBI, two sub-phenotypes have been consistently identified: a hyper-inflammatory sub-phenotype with features such as higher levels of IL-6, IL-8, sTNFR1, higher rates of vasopressor use and lower circulating protein C and bicarbonate than a second hypo-inflammatory sub-phenotype [109]. The two phenotypes have been related to different responses to several therapies and highlighted bicarbonate, IL-6, IL-8, CRP, sTNFR-1 and vasopressor biomarkers as the most predictive variables for ventilator-free days and organ failure-free days.

Regarding septic shock therapy, another randomized trial highlighted data obtained from machine learning that has shown the IFNγ/IL10 ratio to be a good biomarker for the decision to administer hydrocortisone in septic shock [74]. Antibiotic administration and its optimization in critically ill children have also been studied as a field for potential algorithm implementation [110,111]. A recent study has analyzed the impact of a biomarker-based algorithm on broad-spectrum antibiotic prescribing in children with new-onset SIRS without proven bacterial infections admitted in a PICU [110]. This algorithm stated that PICU physicians should consider stopping antibiotics if: sterile site cultures obtained at SIRS onset revealed no growth after 48 h, onset CRP and PCT were low, and there was no sign of infection at the exam or imaging. The authors noted a reduction in excessive broad-spectrum antibiotic therapy after the algorithm implementation in patients in which a bacterial infection had been found, while no differences were seen in the so-called uninfected patients except for the ones who had low biomarkers at the onset [110]. While de-escalation of antibiotic therapy in critically ill children remains a controversial topic, algorithms might ease the decision for patients with low biomarkers.

## 6. Conclusions

Inflammatory response is a complex mechanism in which many biochemical and immunological factors contribute to the host response in SBI. Perfecting biomarker accuracy could be useful for antimicrobial stewardship, pointing to more appropriateness in antibiotic prescription and dosage.

CRP and procalcitonin still represent the most used biomarkers in the pediatric ED for the diagnosis of SBI. Their sensibility and sensitivity increase when combined, and for this reason, it is reasonable to take them both into consideration in the evaluation of febrile children. Omics technologies (i.e., microarrays, next-generation sequencing, microRNAs, metabolomic phenotyping using nuclear magnetic resonance imaging, and mass spectrometry) have recently been used to identify markers of sepsis [10]. The information derived in this regard is presently very poor, and further studies are needed to understand the interactions between genes and biomolecules as well as for their use in daily clinical practice. The potential of machine learning tools, which represent a real novelty in medical practice, in conjunction with routine clinical and biological information, may improve the accuracy of diagnosis and target therapeutic options in SBI. However, studies on this matter are not yet validated in younger populations, making their relevance in pediatric precision medicine still uncertain. More data from further studies are necessary to improve clinical practice and decision making using these new technologies.

## Figures and Tables

**Figure 1 biomolecules-14-00097-f001:**
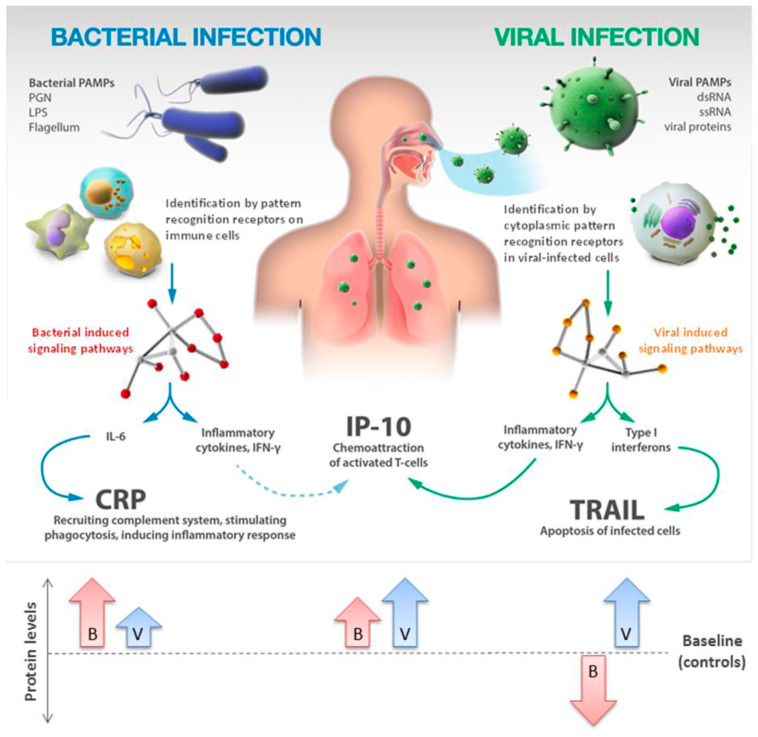
Novel host-immune signature for distinguishing between bacterial and viral infections. Arrows indicate increase or decrease in bacterial (B) and viral (V) infections.

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
