# Peer review of "Biomarkers for Serious Bacterial Infections in Febrile Children"

_biomolecules, 2024, doi:10.3390/biom14010097_

Round 1

Reviewer 1 Report

Comments and Suggestions for Authors

Good review.

Additional comments should be made regarding Figure 2, lines 373-375.

Figure 2 seems redundant.

Include comments on the value of blood cultures and of time to positivity.

Some references that could help:

Blot F. Lancet 1999; 354:1071 (historical value).

McGowan KL. Pediatrics 2000; 106:251.

Alpern ER. Pediatrics 2000; 106:505.

LI Q. Eur J Clin Microbiol Infect Dis 2019; 38:457.

Kang CM. J Microbiol Immunol Infect 2020; 53:882.

She RC. J Appl Lab Med 2019; 3:617.

Matsushita FY. Clinics (Sao Paulo) 2022; 78:100148.

HsiehYC. BMC Infect Dis 2022; 22:142.

References 51 and 26 could also be used.

Comments on the Quality of English Language

Minor editing

Author Response

Good review.

Re: Thank you for your positive evaluation. We further revised the text according to your comments and those received from the other reviewers.

Additional comments should be made regarding Figure 2, lines 373-375. Figure 2 seems redundant.

Re: According to your suggestions, Figure 2 has been deleted. Moreover, according to the comments received from another reviewer, we also deleted Table 1 (p. 4).

Include comments on the value of blood cultures and of time to positivity.

Some references that could help:

Blot F. Lancet 1999; 354:1071 (historical value).

McGowan KL. Pediatrics 2000; 106:251.

Alpern ER. Pediatrics 2000; 106:505.

LI Q. Eur J Clin Microbiol Infect Dis 2019; 38:457.

Kang CM. J Microbiol Immunol Infect 2020; 53:882.

She RC. J Appl Lab Med 2019; 3:617.

Matsushita FY. Clinics (Sao Paulo) 2022; 78:100148.

HsiehYC. BMC Infect Dis 2022; 22:142.

References 51 and 26 could also be used.

Re: Your suggestions on time of blood culture positivity have been considered and all the suggested references added (p. 8).

Reviewer 2 Report

Comments and Suggestions for Authors

Severe bacterial infections (SBI) such as bacterial sepsis are the cause of major health concern, as they are often life threatening, especially in children. Therefore, it’s important for caregivers in the emergency department to be aware of the signs and symptoms of bacterial sepsis in children and seek medical attention promptly if they suspect their child may be affected. Moreover, distinguishing between viral (that do not need require antibiotic treatment) and bacterial infections (that need administration of antibiotics) is very important as early intervention and treatment significantly improve the chances of a positive outcome.

In the presented review, the authors aimed to evaluate the specificity and sensitivity of several, currently used biomarkers to effectively identify SBI in children and neonates presented at emergency department (ED). I would like to suggest the following revisions:

1.    Provide references for the following statements- lines 66-67A larger number of neutrophils are consumed at the site of SBI, and they continue to be supplied to the infected site from the bone marrow via the bloodstream.”, lines 106-107CRP is currently one of the most frequently used biomarkers for infection in the ED worldwide”, line 241-242Many authors had explored the association between soluble TRAIL 241 (sTRAIL) levels in septic patients and the risk of mortality”, the authors say many authors..but have only provided one reference.

2.    In the cytokine section of the review, briefly explain how each cytokine is released (e.g. which TLR-PAMP pair triggers the release of the cytokine and why this is specific for bacteria v/s viruses). Similarly, please include 1-2 lines of explanation for sTREM-2 about how is this molecule released and how is it relevant for bacterial infections.

3.    Explain lines 199-200 “However, the poor predictive accuracy of this molecule doesn’t outperform the discrimination of traditional sepsis biomarkers in the clinical practice.”

4.    The image resolution for Figure 1 is extremely low and it is difficult to read. Please provide an image with better quality and include explanation in the figure legend especially for the bottom part of the figure that involves arrows and abbreviations.

Comments on the Quality of English Language

Editing is required, especially in the later parts of the manuscript (Cytokines and the parts following that section). 

Author Response

Severe bacterial infections (SBI) such as bacterial sepsis are the cause of major health concern, as they are often life threatening, especially in children. Therefore, it’s important for caregivers in the emergency department to be aware of the signs and symptoms of bacterial sepsis in children and seek medical attention promptly if they suspect their child may be affected. Moreover, distinguishing between viral (that do not need require antibiotic treatment) and bacterial infections (that need administration of antibiotics) is very important as early intervention and treatment significantly improve the chances of a positive outcome.

Re: Thank you for your comments. We revised the manuscript as suggested.

In the presented review, the authors aimed to evaluate the specificity and sensitivity of several, currently used biomarkers to effectively identify SBI in children and neonates presented at emergency department (ED). I would like to suggest the following revisions:

  1. Provide references for the following statements- lines 66-67 “A larger number of neutrophils are consumed at the site of SBI, and they continue to be supplied to the infected site from the bone marrow via the bloodstream.”, lines 106-107 “CRP is currently one of the most frequently used biomarkers for infection in the ED worldwide”, line 241-242 “Many authors had explored the association between soluble TRAIL 241 (sTRAIL) levels in septic patients and the risk of mortality”, the authors say many authors..but have only provided one reference.

Re: References have been added as suggested (pp. 2, 3 and 7).

  1. In the cytokine section of the review, briefly explain how each cytokine is released (e.g. which TLR-PAMP pair triggers the release of the cytokine and why this is specific for bacteria v/s viruses). Similarly, please include 1-2 lines of explanation for sTREM-2 about how is this molecule released and how is it relevant for bacterial infections.

Re: Revised as suggested (pp. 5 and 8).

  1. Explain lines 199-200 “However, the poor predictive accuracy of this molecule doesn’t outperform the discrimination of traditional sepsis biomarkers in the clinical practice.”

Re: The text has been clarified (p. 6).

  1. The image resolution for Figure 1 is extremely low and it is difficult to read. Please provide an image with better quality and include explanation in the figure legend especially for the bottom part of the figure that involves arrows and abbreviations.

Re: We can provide the original image in ppt if needed by the Editor. A legend has been added (p. 7).

Reviewer 3 Report

Comments and Suggestions for Authors

Dear Authors, 

Thank you for sending your manuscript. Unfortunately, your manuscript is not comprehensive enough to be published in our journal. For examples: 

1. Why did you search the articles that have been published just for the last 5 years? 

2. There are some missing Mesh terms in your PubMed search, such as neonatal sepsis 

3. Your title mentions predictor models, but you do not explain them enough in the manuscripts. 

4. Table 1 is not enough. It would be best if you created a more comprehensive table. 

5. There is no explanation for Figure 1 and Figure 2; you do not refer to them in the manuscript. 

6. There is no information about transcriptomics and mass spectrometry. 

7. Maybe you should mention overlapping diseases such as viral and bacterial infection together: RSV and bacterial infection 

I recommend working on this manuscript further. Good luck! 

Best regard

Comments on the Quality of English Language

Dear Authors, 

Thank you for sending your manuscript. I believe that it would be better to edit the English quality of the manuscript. For examples: 

1. Line 181: wasn’t - was not. 

2. Line 199: doesn’t - does not 

3. Line 245: It cannot be started in sentences with an abbreviation. 

Best, 

Author Response

Dear Authors,

Thank you for sending your manuscript. Unfortunately, your manuscript is not comprehensive enough to be published in our journal.

Re: Thank you for your comments. We revised our manuscript according to your suggestions.

For examples:

  1. Why did you search the articles that have been published just for the last 5 years?

Re: We Guidelines and position papers published from 2003 to 2018 were also considered (p. 2).

  1. There are some missing Mesh terms in your PubMed search, such as neonatal sepsis

Re: We clarified that we did not include manuscripts on biomarkers in neonatal sepsis because we have al-ready analysed this topic in a previous publication (p. 2).

  1. Your title mentions predictor models, but you do not explain them enough in the manuscripts.

Re: The title has been revised (p. 1).

  1. Table 1 is not enough. It would be best if you created a more comprehensive table.

Re: Table 1 has been deleted according to your request (p. 4).

  1. There is no explanation for Figure 1 and Figure 2; you do not refer to them in the manuscript.

Re: Figure 1 was already introduced in the text, we highlighted it in green (p. 7). Figure 2 has been deleted according to reviewer #1 suggestion (p. 10).

  1. There is no information about transcriptomics and mass spectrometry.

Re: A comment on this has been added (p. 10).

  1. Maybe you should mention overlapping diseases such as viral and bacterial infection together: RSV and bacterial infection

Re: We mentioned this and we added three new references (pp. 3 and 11).

I recommend working on this manuscript further. Good luck!

Re: Thank you for your suggestions. We further revised the text according to comments received from you and other three reviewers. We hope that you could accept the manuscript in its revised form.

Round 2

Reviewer 2 Report

Comments and Suggestions for Authors

I am satisfied with the author's responses to my comments. The manuscript can be accepted for publication now. 

Comments on the Quality of English Language

Minor editing is required 

Reviewer 3 Report

Comments and Suggestions for Authors

Dear Authors, 

Thank you for your effort to make the manuscript better. I am happy to write it would be a great asset for our journal. 

Best regard